# Humanized Mice as an Effective Evaluation System for Peptide Vaccines and Immune Checkpoint Inhibitors

**DOI:** 10.3390/ijms20246337

**Published:** 2019-12-16

**Authors:** Yoshie Kametani, Yusuke Ohno, Shino Ohshima, Banri Tsuda, Atsushi Yasuda, Toshiro Seki, Ryoji Ito, Yutaka Tokuda

**Affiliations:** 1Department of Molecular Life Science, Division of Basic Medical Science, Tokai University School of Medicine; 143 Shimokasuya, Isehara-shi, Kanagawa 259-1193, Japan; y-ohno@tsc.u-tokai.ac.jp (Y.O.); shino-w@tokai-u.jp (S.O.); 2Institute of Advanced Biosciences, Tokai University, 4-1-1 Kitakaname, Hiratsuka-shi, Kanagawa 259-1292, Japan; 3Department of Breast and Endocrine Surgery, Tokai University School of Medicine, 143 Shimokasuya, Isehara-shi, Kanagawa 259-1193, Japan; banri@is.icc.u-tokai.ac.jp (B.T.); tokuda@is.icc.u-tokai.ac.jp (Y.T.); 4Department of Internal Medicine, Division of Nephrology, Endocrinology and Metabolism, Tokai University School of Medicine, 143 Shimokasuya, Isehara-shi, Kanagawa 259-1193, Japan; yasuda1633@yahoo.co.jp (A.Y.); tsekimdpdd@tokai-u.jp (T.S.); 5Central Institute for Experimental Animals, 3-25-12 Tonomachi, Kawasaki-ku, Kawasaki-shi, Kanagawa 210-0821, Japan; rito@ciea.or.jp

**Keywords:** peptide vaccine, immune checkpoint inhibitor, humanized mouse, cancer antigen, immune suppression

## Abstract

Peptide vaccination was developed for the prevention and therapy of acute and chronic infectious diseases and cancer. However, vaccine development is challenging, because the patient immune system requires the appropriate human leukocyte antigen (HLA) recognition with the peptide. Moreover, antigens sometimes induce a low response, even if the peptide is presented by antigen-presenting cells and T cells recognize it. This is because the patient immunity is dampened or restricted by environmental factors. Even if the immune system responds appropriately, newly-developed immune checkpoint inhibitors (ICIs), which are used to increase the immune response against cancer, make the immune environment more complex. The ICIs may activate T cells, although the ratio of responsive patients is not high. However, the vaccine may induce some immune adverse effects in the presence of ICIs. Therefore, a system is needed to predict such risks. Humanized mouse systems possessing human immune cells have been developed to examine human immunity in vivo. One of the systems which uses transplanted human peripheral blood mononuclear cells (PBMCs) may become a new diagnosis strategy. Various humanized mouse systems are being developed and will become good tools for the prediction of antibody response and immune adverse effects.

## 1. Introduction

Peptide vaccines are widely accepted as a promising strategy to fight infectious disease and cancer. However, the efficacy of a peptide vaccine depends not only on the antigen presentation through antigen-presenting cells but also on the immune environment of each patient, since the immunity of patients with chronic infectious disease and/or cancers tend to be dampened. Therefore, to achieve a more personalized medicine, we need a more detailed diagnosis before treatment. We propose the use of the humanized mouse system established through transplanting human peripheral blood mononuclear cells (PBMCs) from a patient into an immunodeficient mouse, for the evaluation of the response to peptide vaccines and other reagents which influence patient immunity. We also describe the immune condition artificially induced by immune checkpoint inhibitors (ICIs) [1] and the reagents against immune-related adverse events (irAEs), followed by the current state-of-the-art advances of humanized mouse systems and the issues to overcome. Moreover, we will discuss whether it is possible to evaluate the patient immunity by using second-generation humanized mice.

## 2. Difficulties in the Development of Peptide Vaccines 

The design of peptide vaccines relies on the potential of peptides to bind to the major histocompatibility complex (MHC) in order to be presented by antigen-presenting cells (APCs), such as dendritic cells (DC) and macrophages. However, the MHC binding affinity is not enough to predict the activation of immunity, because the immune condition is different among different patients. Therefore, the decrease in the immune competence should be evaluated when the vaccine is adopted for patients with cancer and/or affected by a chronic infection. The vaccine is not restricted to be used as an anticancer agent; it also includes the influenza vaccine, to be administered to cancer patients [2,3,4]. Moreover, if the immune checkpoint inhibitors (ICIs) are used for the purpose of immune activation, the situation becomes more complex. We discuss the factors in detail below. 

### 2.1. Selection of the Adequate Peptide for Vaccination 

Vaccines are categorized as preventive or therapeutic based on their function and are further classified into virus, peptide, DNA, or DC vaccines, depending on the antigen source. Various types of antigens and adjuvants have been developed and evaluated for vaccination against infectious diseases. The design of the peptide antigen is important for inducing the most effective output with each type of vaccine, as each pathogen has a unique strategy for infection and proliferation. However, for long-lasting memory production, protein/peptide-based antigens are essential because the memory requires the activation of T cells through antigen-presenting cells, such as DC and macrophages. On the other hand, antigens need to activate B cells by crosslinking B-cell receptors (surface Igs). Therefore, the antigen epitope should be exposed to the hydrophilic surface by protruding into the aqueous solution and, thus, being recognized by B cells in vivo.

For the vaccine components to activate T cells, the antigens should at least contain a highly immunogenic peptide with more than 8, and up to 30, residues which can be further presented by the patient MHC (class I for cytotoxic T-cell activation and class II for antibody production). Moreover, as the peptide sequence mutates easily within the virus, it should be selected to maintain the peptide primary structure. The peptide presentation is predicted for HLA and mouse MHC by using available algorithms [5,6,7]. However, the prediction is incomplete because more new HLA types have been reported [8,9,10], and even if a peptide is successfully presented by mouse MHC in an experimental design, it does not imply that the same peptide will be presented on HLA. Therefore, larger peptide antigens are typically used in order to include as many epitopes as possible to be presented by major HLAs. 

The evaluation of adjuvants is also very important. The induction of inflammation by the adjuvant is effective for the enhancement of the immune response. However, inflammation induction may pose a risk and result in adverse effects for patients. Therefore, self-adjuvanting techniques have been developed for clinical use [11,12,13,14]. Among them, the conjugation of molecules related to the ligands of tool-like receptors (TLR) to target peptides may be a safe and effective vaccine adjuvant. The DNA vaccines now, on translational research, use genes of TLR-related molecules.

### 2.2. Antigens which Enable Activation of the Patient Immune System

While vaccination is the most effective strategy to prevent acute infectious diseases caused by bacteria and viruses, it is not easy to develop effective vaccines against cancer and chronic infectious diseases. Similarly, to antigens present in pathogenic bacteria and viruses, patients with cancer present tumor-associated antigens (TAA) with high antigenicity and immunogenicity. TAAs are classified into differentiation, tissue-specific, mutated, and overexpressed antigens [15]. The U.S. Food and Drug Administration (FDA) has already approved for clinical use several cancer vaccines based on TAAs [16,17]. Hepatitis B virus (HBV) and human papilloma virus (HPV) are examples of TAA-based vaccines [18]. There are also unique classic vaccines like sipuleucel-T, the first therapeutic cancer vaccine approved by the FDA [19]. Moreover, many cancer vaccine candidates are currently under investigation in clinical trials, including nucleic acid-containing liposomes and nanoparticles (DNA vaccines) and gp100 peptide (peptide vaccines) [20,21,22].

On the other hand, especially for tumor-associated peptide vaccines, even if the antigen presentation is satisfied, it is difficult to activate the patient immune system. In spite of the extensive development of vaccines which may induce an anticancer immune response in patients, this response may vary among patients, making the vaccine not always effective. The immune-reactive tumors are called hot tumors, whereas nonimmune reactive tumors are referred to as cold tumors. Hot tumors are thought to have much more cell mutations compared to cold tumors, suggesting that hot tumors have many more TAAs [23]. Accordingly, the hot tumor, which is immune-reactive for the patient, may become the target of peptide vaccines, whereas, in the case of nonreactive cold tumors, the peptide vaccine might be ineffective. In hot tumors, there are some antigens that are highly expressed because of their overexpression on cancer cells. Human epidermal growth factor receptor 2 (HER2) is an example of a TAA molecule, as HER2 is overexpressed on the tumors of patients with breast cancer, and the specific antibody Herceptin is very effective for suppressing cancer progression. Due to the success of antibody reagents, many other human antitumor IgGs have been developed, and their mechanism of action has been investigated [24]. However, the antitumor effect does not last long enough, and the mechanism underlying this effect has not been fully elucidated.

Another problem in the development of cancer vaccines is the incomplete prediction resulting from the algorithms used. Our immune system rejects self-antigen-reactive clones, which may contain cancer-specific clones. Therefore, many of the predicted peptides cannot induce the desired immune response, even though the peptide leads to an immune response in experimental animals. Even if the peptide functions as an antigen, cancer cells have heterogenous mutations in the tumor mass, and, thus, the reactivity of each cancer cell is predicted to be diverse. Therefore, a complete rejection of the cancer cells within the tumor mass is difficult if simply one TAA is selected as peptide antigen.

### 2.3. Immune Suppression in Patients Prevents the Effectiveness of Vaccines

The most important challenge in the design of a vaccine is the immune suppression caused by the patient. The levels of cytotoxic T-cell activation, antibody production, and productive inflammation are different among patients with cancer. Therefore, we cannot predict the patient immune response, even if the peptide vaccine induces an immune response that is similar to the one produced by a viral infection in a healthy individual.

Therefore, although peptide vaccines have been extensively developed, the effect of the anticancer peptide vaccine is very limited, even if the peptide is presented by class I HLA on the patient DCs and the beneficial effect remains, as reviewed by Wong et al. [25]. One of the reasons for this limited effect is that cancer cells are originally “self”, and the immune response is basically suppressed by clonal deletion or regulatory immune cell reactions, even though the peptide-reactive CD8 T cells are often detected in the patient PBMC. Even if mutations occur, most of them are limited to a very small region, and the peptides recognized as “non-self” might be very few or suppressed. This mechanism is present in cold tumors.

Meanwhile, an autoimmune disease might be induced by the suppression of peripheral tolerance. The neutrophil extracellular traps (NETs) play a role in the development of autoimmunity [26,27]. NETs are networks of extracellular fibers that are primarily composed of DNA from neutrophils, which suppress the movement of pathogens. Neutrophils release granule proteins, together with chromatins, and form an extracellular fibril matrix of NETs. The autoantigens involved in neutrophil granular proteins contain very common proteins, such as actin and histones. The proteins vary with the stimulation, and they occasionally induce an autoimmune response. It is important to understand which condition determines if the immune system will or will not induce an autoimmune disease. Moreover, not only cancers, but also some pathogens, induce tolerance. Actually, immature DCs, which induce only an MHC-TCR signal, may induce anergy to self-reactive and non-self-reactive T cells [28]. 

## 3. Immune Checkpoint Inhibitors and Reagents for Side-Effect Regulation

Recently, adaptive immune-resistant tumor cells which express the programmed-death-L1 (PD-L1) antigen were reported in melanomas by Abiko et al. [29] and Taube et al. [30]. According to their reports, PD-L1 is largely induced on the local tumor cells by tumor-infiltrating lymphocytes (TILs)-derived IFN-γ because IFN-γ is the most potent inducer of PD-L1in inflammatory cytokines. Upregulation of PD-L1 by IFN-γ has been extensively described in various cell types [31,32,33,34,35,36,37]. Similarly, TNF-α, another pro-inflammatory cytokine, also upregulates PD-L1 expression via TNF-α-NF-κB pathway [38,39,40]. TNF-α is reported to synergistically act with IFN-γ to induce PD-L1 expression at both mRNA and protein levels. IFN-γ enhances the resistance of the adaptive immune response by PD-L1 induction in hepatocellular carcinoma cells which upregulate the expression of IFN-γ receptors [41]. PD-L1 is expressed not only in all hematopoietic cells but also in many non-hematopoietic cell types, such as endothelial and epithelial cells [42,43]. In contrast, PD-L2 expression is more restricted to professional antigen-presenting cells, such as DCs, B cells, and monocytes/macrophages. Besides PD-1, there are other known interacting partners for PD-L1 and PD-L2. PD-L1 also binds to CD80, whereas PD-L2 uses repulsive guidance molecule (RGM) domain family member B (RGMB) as an alternative binding partner. Both types of interaction also inhibit immune responses [44,45].

### 3.1. Patients with Cancer 

Recently, the anticancer effect of various immune checkpoint antibodies was elucidated [46]. The “immune checkpoint antibody” induces the blockage of continuous T-cell activation in the periphery. PD-1 antigen is expressed on the long-lived activated T cells, exhausted T cells, and the follicular helper T cells (Tfh) [47,48]. Normally, PD-L1 is expressed on antigen-presenting cells and germinal center B cells [49,50]. Apoptosis is induced when the PD-1-expressing T cells encounter the PD-L1-expressing APCs [49]. When the PD-1/PD-L1 interaction is inhibited by the anti-PD-1 antibody, T cells survive, and the anticancer effect is prolonged. Other immune checkpoint molecules, such as CTLA-4, PD-1, TIM-3, and LAG-3, have been reported, and the ability of the antibodies against such immune checkpoint molecules is being evaluated as anticancer products [51,52]. The effect is remarkable, but the response is still limited to a fraction of patients with cancer. The effect is ordinary, not long-lasting, and the combination of these inhibitors and other anticancer drugs are under investigation. 

Moreover, antibodies are so expensive that, before using them as therapeutic agents, a strategy is needed to distinguish among patients that are responsive to the treatment from those that are not. Many biomarkers have been reported to predict the efficacy of the treatment. However, the heterogeneity of tumor masses and the variety of antibodies available make it difficult to find such predictive biomarkers, and even PD-L1 expression might not be a promising marker. Collectively, many studies have suggested that PD-L1 expression on melanoma cells can represent a biomarker to test for the efficacy of anti-PD1 and related antibodies, such as Nivolumab, Ipilimumab, and Pembrolizumab [53,54,55], and other immune checkpoint inhibitors; however the PD-L1 expression is not always an effective marker for patients with cancer in other clinical trials [56,57]. For example, PD-L1 expression on melanoma cells in pretreatment tumor biopsy samples is reported to correlate with response rate, progression free survival, and overall survival in patients with advanced melanoma treated with anti-PD1 antibodies [55], but these antibodies are also effective for PD-L1-negative patients [57].

While the benefits of assessing PD-L1 expression on melanoma cells to predict the clinical outcomes of ICI.

It is already defined. treatment have been suggested, as above, there are still no common criteria of diagnosis. This fact limits the clinical usefulness of the diagnosis of PD-L1 expression, because the low sensitivity of immunohistochemical (IHC) assays using different antibody clones makes it difficult to establish staining platforms and scoring systems [54,55,57,58,59]. To avoid misprediction by IHC staining, Conroy et al. assessed the expression of PD-L1, using next-generation RNA sequencing, but the sensitivity of their system resembles that of IHC assay systems and is, in addition, more expensive [58]. Additional assays or completely different assay systems will be needed in the future to diagnose PD-L1 expression of patient cancer tissues, for the prediction of clinical outcomes for the ICI treatment of melanoma [60].

### 3.2. Patients with Infectious Diseases

Viral infections do not always enhance PD-L1 expression, because similar PD-L1 levels are detected in individuals not infected with viruses [61,62,63,64]. Increased PD-L1 levels are related to specific viruses, such as the following: Epstein–Barr virus (EBV) [65,66,67,68], hepatitis B virus (HBV) [69,70,71], hepatitis C virus (HCV) [72,73,74,75], human immunodeficiency virus (HIV) [63,76,77,78,79], human papilloma virus (HPV) [68,80,81,82,83], Merkel cell polyomavirus (MCPyV) [84], bovine leukemia virus (BLV) [85], and Kaposi sarcoma-associated herpes virus (KSHV) [86]. The pathobiological mechanisms by which viruses trigger the expression of PD-L1 have been elucidated. Pathogen-associated molecular patterns (PAMPs) such as lipopolysaccharides (LPS), double-stranded RNA, and non-methylated CpG, from virus, bacteria, and fungi, activate toll-like receptors (TLRs) to induce the immune response and protect the host against the infection. Therefore, the effect of PD-1/PD-L1 blockage by ICI might not be limited to blocking cancer-T-cell interaction. Other hematopoietic lineage cells expressing PD-1 and/or PD-L1 might also be affected. For example, a fraction of plasmablasts and regulatory B (Breg) cells also express PD-L1 [87,88]. Therefore, the blockage of the axis may affect the humoral immunity or Breg cells. However, the antigen-specific reaction in such a systemic immunity is difficult to analyze in vivo.

### 3.3. Steroid Hormones and ICI Side Effects

Glucocorticoids are a class of steroid hormones that are powerful immune-suppressants that produce an effect on the systemic immune response. Conditions such as pregnancy and chronic inflammation may induce glucocorticoid secretion. Glucocorticoids [89] secreted by the stimulation of chronic inflammation are widely used as anti-inflammatory drugs. While they induce various signals related to cytokine and Fc receptors that modify metabolism and immune responses, it was recently reported that glucocorticoids impair upstream B-cell-receptor and Toll-like-receptor 7 signaling, reduce transcriptional output from the immunoglobulin loci, and promote significant upregulation of genes encoding the immunomodulatory cytokine IL-10 and the terminal-differentiation factor BLI MP-1 [90]. Expression of κ light chain and the two variable regions are especially suppressed. If patients affected with cancer or severe infectious diseases increase their glucocorticoid levels in order to overcome the disease-induced inflammation, or if they are treated with glucocorticoid because of the regulation of anticancer drug-induced side effects, the anticancer Ig expression might be suppressed. If the inflammatory, glucocorticoid-abundant condition continues, the potential for antibody production in the patient may be dampened. Therefore, if the PBMC of patients is examined for the antibody-production response, we may be able to predict if the patient is exposed to such steroid-based immune suppression. Glucocorticoids have also been reported to enhance metastasis in breast cancer [91]; therefore, their effect on patients with cancer needs to be examined in more detail.

On the other hand, it has been reported that ICI treatments occasionally induce a typical side effect related to pituitary dysfunction. Notably, hypophysitis, a previously very rare disease, has emerged as a distinctive side effect of ipilimumab and occasionally of nivolumab [92]. These side effects are not limited only to the pituitary; they also affect the thyroid, adrenal glands, and other downstream-target organs [93].

## 4. Humanized Mouse Models for the Evaluation of the Human Immune Environment

As we mentioned above, the prediction of the protective immunity development by vaccination is difficult because the immune condition is diverse in each patient, and the appropriate ICIs and induced irAEs may not be predicted. In order to determine the protocol reflecting the immune condition of each patient, the so-called personalized medicine, a humanized mouse system reconstituted with the patient immunity, may be useful [94]. The immunization with vaccines may reveal not only the effect of a specific vaccine, but it may also provide information regarding the patient immune response to mimic the anticancer/pathogen response. The current status of the humanized mouse system involving next generation humanized mice and its limitations is shown in Figure 1 (cellular immunity) and discussed below [95,96].

### 4.1. Humanized Mice for Reconstitution of the Human Immune System with Hematopoietic Cells

The humanized mouse system was originally developed to evaluate the multipotency of human HSCs or progenitors. Severely immunodeficient mouse strains, as well as the transplantation techniques, have recently been developed [97,98,99,100]. After the discovery of the nonobese diabetic severe combined immunodeficient mouse (NOD-scid) model and its derivatives, transplantation of human hematopoietic stem cells (HSCs) into these mice led to the development of human lymphocytes and myeloid cells which, are localized in the primary and secondary lymphoid tissues of the mouse [101,102,103]. These mouse models have been used to analyze the differentiation of human hematopoietic and leukemic stem cells [104]. On the other hand, because of the success of humanized monoclonal antibody reagents such as trastuzumab and rituximab, completely human-type antibody production has also been attempted, using these mouse models transplanted with various types of human hematopoietic cells [105].

NOD/Shi-scid-IL2Rγ^null^ (NOG), developed at the Central Institute for Experimental Animals, and NOD scid gamma (NSG), developed at the Jackson Laboratory, are two representatives of severely immunodeficient mouse strains. Both mouse strains have a deficiency in IL-2rgc [97,106,107,108]. NOG mice possess a truncated IL-2rgc, and NSG mice have a complete deletion of the gene coding for IL-2rgc; the efficiency of the engraftment and the differentiation efficiency are comparable in the two strains. Both of them enabled the development of human T and B cells from human HSCs in a xenogenic environment. However, most of the human B cells differentiated in the mice expressed CD5, a marker of B1 cells, and the specific IgG antibody is not produced [109,110,111,112,113] (Table 1). We reconstructed human immunity in NOG mice transplanted with HSC and immunized with CH401MAP, a specific HER2 peptide antigen for patients with breast cancer, and keyhole limpet hemocyanin (KLH), or toxic shock syndrome toxin-1 (TSST-1), with Freund’s complete adjuvant and measured the specific antibody titer by ELISA. As a result, although antigen-specific IgM and nonspecific IgG were detected in the sera, antigen-specific IgG was not detected in mice (Table 1) [114,115,116]. These mice did not develop a germinal center, which has a structure composed of T, B, and follicular DCs and plays a crucial role in highly specific crass-switched IgG antibody production. The results indicated that human T cells and B cells developed in the mouse environment could not induce cognate interaction, because the T cells are selected for mouse MHC in the thymus.

After the first trial with NOG and NSG mouse models, the animals with mouse MHC knockout and HLA transgenic antigen were developed to induce cognate interaction of T cells and B cells. Among them, HLA class I transgenic mice evoked antigen-specific cytotoxic T-cell response against HSV virion protein peptide [128] or WT1 peptide [129]. The success of the reconstitution of human cellular immune response was followed by an adoptive transfer therapy model using the humanized-mouse system [130]. Consequently, the established patient-derived xenograft (PDX) system, which transplants a patient’s cancer tissues (minimal standard was reported by Meehan et al. [131]), combined with a patient’s T cells, is widely accepted. The detail was intensively reviewed by other researchers [132,133,134].

On the other hand, the response of HLA class II transgenic mice did not completely mimic the human humoral immunity [119,120,125]. Moreover, mice need to be transplanted with the same HLA-bearing human HSCs, which restrict the samples to be examined. Among them, Ashizawa et al. reported that class I and class II MHC KO NOG mice (NOG dKO) transplanted with human PBMC and tumor cell lines showed higher anticancer effects after PD-1 antibody treatment [135]. In these mouse strains, transplanted tumor cells and immune cells can be engrafted, and the anticancer effect of human immune cells can be observed (reviewed by Chen et al. [95]). The mouse system had an advantage, which is that the restriction of HLA type could be avoided by using PBMC, which contain the same patient’s T cells and antigen-presenting cells. However, they did not detect anticancer antibody production in this study.

Currently, various transgenic mouse strains expressing human cytokines and surface antigens, along with more severely immunodeficient mouse strains, are being developed to transplant human hematopoietic cells (HSC or PBMC). The category of newly-established mouse system includes myeloid cell development, cancer immunotherapy model, allergy model, and graft-versus-host disease (GVHD) model.

Another humanized mouse model, called BLT mice, has been reported. In this mouse model, immunodeficient mice are co-transplanted with human fetal liver and thymus tissues, along with autologous CD34 + HSCs. This mouse system is a modification of the SCID-Hu mice developed by MacCune [113,117,123]. In these mice, antigen-specific antibody production was partially achieved, and experiments on infection with bacteria or viruses were conducted [118]. Severely immunodeficient NSG mice are used to establish NSG–BLT mice [136]. A modified NSG mice, in which Human *SCF*, *GM-CSF*, and *IL-3* genes were transduced, was used to establish an improved BLT mouse strain. Based on the NSG mouse strain, human HSCs, fetal liver, and fetal thymus were transplanted, and mice were inoculated with dengue and/or Zika virus. As a result, these mice induced a higher immune response than that of conventional NSG mice, although graft-versus-host disease (GVHD) could not be avoided [124,126,127]. However, because of a serious ethical problem, Japanese researchers are unable to establish the BLT mouse system. The BLT model system succeeded in the induction of the cytotoxic immune response with no mature humoral immunity, maybe because the cytotoxicity is too high to maintain the antibody production (discussed in [94]).

Collectively, many of the strains support the differentiation of various hematopoietic cell lineages from human HSCs. Moreover, PBMC engrafts in the mice and can reconstitute human cellular immunity. However, human humoral immune response in a mouse model still needs further improvement: it is impossible, so far, to reconstruct the immune condition involving humoral immunity of various patients.

### 4.2. Humanized Mouse System to Evaluate Antigen-Specific Antibody Production

It is difficult to completely develop humoral immunity in humanized mice because of the reasons exposed above. While T cell–B cell interaction needs cognate interaction, humans have a large variety of HLA types, and it is difficult to cover all the HLA types present in a patient blood. Immunodeficient mice transplanted with PBMCs are promising tools to evaluate human immune responses to vaccines, compared to the HSC-transplanting mouse system. However, these mice usually develop severe GVHD [137]. With GVHD, mice develop a large amount of activated T cells, while B cells are decreased in parallel, and there is no humoral immune response. Therefore, it is difficult to evaluate the production of antigen-specific IgG production after antigen immunization in those mice. To evaluate antigen-specific IgG responses in PBMC-transplanted immunodeficient mice, we developed a novel NOD/Shi-scid-IL2rg^null^ (NOG) mouse strain that systemically expresses the human IL-4 gene (NOG-hIL-4-Tg) [116]. After human PBMC transplantation, GVHD symptoms were significantly suppressed in the Tg NOG, as compared to conventional NOG mice. In the kinetic analyses of human leukocytes, long-term engraftment of human T cells has been observed in peripheral blood of NOG-hIL-4-Tg, and then CD4+ T cells dominantly proliferated rather than CD8+ T cells. Furthermore, these CD4+ T cells produced large amounts of IL-4 but suppressed IFN-g expression, resulting in long-term suppression of GVHD. Most of the human B cells detected in the transplanted mice showed a plasmablast phenotype. Vaccination with HER2 multiple antigen peptide (CH401MAP) or keyhole limpet hemocyanin (KLH) successfully induced antigen-specific IgG production in PBMC-transplanted NOG-hIL-4-Tg. The HLA haplotype of donor PBMC might not be relevant to the ability of an antibody secretion after immunization. The reason why NOG-hIL-4-Tg retain B cells and succeeded in the specific antibody production was examined, and we found that the engrafted human lymphocytes decreased glucocorticoid receptor expression, which dampens the humoral immunity [138].

This evidence suggests that the PBMC-transplanted NOG-hIL-4-Tg mouse system is an effective tool to evaluate the production of antigen-specific IgG antibodies, following vaccination in individual cancer patients [116]. The mouse system can be used for the evaluation of the effect of ICIs on antibody production in the presence of human PBMCs, as well.

Of course, the vaccination is not limited to cancer vaccines. As plasmablasts are efficiently developed, the evaluation of vaccines against highly deleterious pathogens, such as Ebola virus, may become possible. Moreover, the donors recovered from such serious infectious disease may keep their memory B cells against the pathogen. Therefore, the transplantation of the PBMCs may develop plasma cells that secrete effective antipathogen antibodies. If we establish the technology for monoclonal antibody preparation, we may obtain the monoclonal antibody reagents for the treatment of such deleterious infectious diseases.

The humanized mouse systems discussed are summarized in Table 1.

## 5. Future Perspectives

Because the efficacy of the peptide vaccine is influenced by the immune-cell environment and the patient’s body fluid content, we need to evaluate vaccines by constructing patient-mimicking conditions. If we can establish patient-PBMC-based check systems using the humanized mouse model for vaccination and additional reagents, we may check the vaccination efficiency, ICI, and IAEs at the same time. If those goals are achieved, they may enable a promising personalized medicine, such as in the case of the use of the mixed lymphocyte reaction for blood-type examination before transplantation. Therefore, it is urgent to develop humanized mice which reconstitute not only human immune cells but the environment of the actual patient. By using the PBMC-based humanized mouse system, various vaccines can be evaluated for their efficacy. We need to improve the humanized mouse system to fine-tune the peptide design for vaccine development.

## Figures and Tables

**Figure 1 ijms-20-06337-f001:**
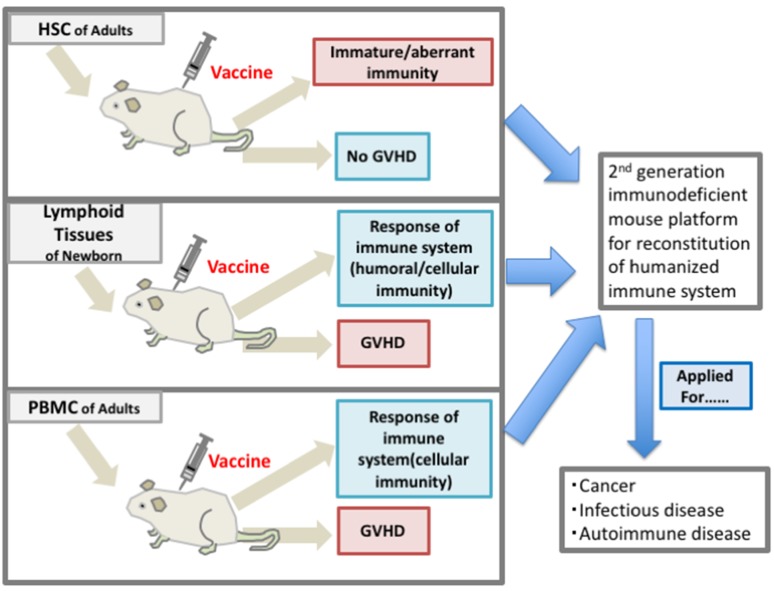
Three strategies for the reconstitution of human immunity in the immunodeficient mouse. The transplanted tissues are HSC, Lymphoid tissues or the fragments of mnewborn, and PBMCs. Many kinds of antigens and pathogens were used for the analysis.

**Table 1 ijms-20-06337-t001:** Humanized mice with antigen-specific antibody production.

Mouse Strain.		Transplanted Tissues	Antigen	Isotype	Reference
**SCID-Hu**	SCID	human fetal liver and thymic fragments under kidney capsule	pneumococcal vaccine	IgG	McCune JM 1988 [117], Aaberge IS et al., 1992 [118]
Hu-HSC	NOG	HSC(CB/MPB/BM) i.v.	DNP-KLH/CH401MAP/TSST-1	IgM	Matsumura T et al., 2003 [109], Kametani Y et al., 2006 [114]
	NSG; Balb/c-Rag1(-/-) gammac(-/-); C.B-17-scid/bg	HSC(CB/MPB/HFL) i.v.	KLH/inactivated H5N1 influenza virus	IgM, IgG	Lepus CM et al., 2009 [111]
	NOG	CD34 + HSC i.v.	OVA	Igs	Yajima M et al., 2008 [98], Watanabe Y et al., 2009 [112]
	NSG	human CD34 + HSC i.v.	OVA, HIV	IgM, IgG	Wtanabe S et al., 2007 [110], Singh M et al., 2012 [102]
	NOG-HLA-DR4/Ab KO	human CD34 + HSC i.v.	OVA	IgM, IgG	Suzuki M et al., 2012 [119]
	NSG-HIS-CD4/B	human CD34 + HSC i.v.	Plasmodium falciparum, circumsporo-zoite (PFCS) protein	IgG	Huang J et al., 2015 [120]
Hu-PBL	SCID	human PBMC i.v.	xenograft	IgM, IgG	Williams S et al., 1992 [121]
	NOG-IL-4-Tg	human PBMC i.v.	KLH/CH401MAP	IgG	Kametani Y et al., 2017 [116]
	DKO-NOG	human PBMC i.v.	human Liver xenograft,	Igs	Aono S et al. 2018 [122]
BLT	SCID	human fetal liver and thymic fragments under kidney capsule with autologous CD34 + HSC		IgG	McCune JM et al., 1988 [123], Aaberge IS et al., 1992 [118]
	NOD-SCID	human fetal liver and thymic fragments under kidney capsule with autologous CD34 + HSC	HIV-1, WNV envelope protein	IgM, IgG	Biswas et al., 2011 [113]
	NSG	human fetal liver and thymic fragments under kidney capsule with autologous CD34 + HSC	pneumococcal vaccine, Dengue virus infection, Zika virus, HIV -1 gp120	IgM, IgG, IgA	Jaiswal S et al., 2015 [124], Jangalwe S et al., 2016 [125], Schmitt K et al., 2018 [126], Gawron MA et al., 2019 [127]

Representative immune-humanized mouse systems which induced antibody production are shown. The data are based on PubMed, published from 1988 to 2019.

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
