# Peer review of "Humanized Mice as an Effective Evaluation System for Peptide Vaccines and Immune Checkpoint Inhibitors"

_ijms, 2019, doi:10.3390/ijms20246337_

Round 1

Reviewer 1 Report

The review article authored by Dr. Tokuda et al. described the importance of using the humanized mouse model for evaluating the success of different immunotherapy such as cancer vaccine and checkpoint inhibitors. The authors described the difficulties and the reasons behind limited success of those immunotherapies clinically. However, there are some important points that need to be addressed:

The authors did not mention other immunotherapeutic approaches such as adoptive transfer therapy as that was used already in humanized mouse system. Ref: https://www.ncbi.nlm.nih.gov/pmc/articles/PMC5617838/ In table 1, search criteria need to specify such as whether Pubmed or other sites were utilized and which date range was used. Even though the mechanism of generating humanized mice were discussed, in order to draw reader's attention, a graphical figure is also necessary especially for non-expert readers. Formatting needs to be corrected in page 6, line 253-254.

Author Response

Response to the comments of Reviewer #1

Comment 1: The authors did not mention other immunotherapeutic approaches such as adoptive transfer therapy as that was used already in humanized mouse system. Ref: https://www.ncbi.nlm.nih.gov/pmc/articles/PMC5617838/

Response: We appreciate the reviewer’s suggestion with reference. We added the description with the reconstitution of cellular immune response achieved by humanized mouse as “Among them, HLA class I transgenic mice evoked antigen-specific cytotoxic T cell response against HSV virion protein peptide [130] or WT1 peptide [131]. The success of the reconstitution of human cellular immune response is followed by an adoptive transfer therapy model using humanized mouse system [132]. Consequently, the established patient-derived xenograft (PDX) system which transplants patient’s cancer tissues (minimal standard was reported by Meehan et al. [133]) combined with patient’s T cells is widely accepted. The detail was intensively reviewed by other researchers [134-136].” In page 9 line 293-301.

Comment 2: In table 1, search criteria need to specify such as whether Pubmed or other sites were utilized and which date range was used.

Response: We appreciate the reviewer’s comment. According to the comment, we added the table legend “Representative immune -humanized mouse systems  which induced antibody production are shown. The data are based on PubMed published from 1988 to 2019.” in page 8 line 288-289

Comment 3: Even though the mechanism of generating humanized mice were discussed, in order to draw reader's attention, a graphical figure is also necessary especially for non-expert readers.

Response: We appreciate the reviewer’s suggestion. According to the suggestion, we added a Figure which explain the humanized mouse model in page 7 after line 247 with figure legend “Figure 1. Three strategies for the reconstitution of human immunity in the immunodeficient mouse. The transplanted tissues are, HSC, Lymphoid tissues or the fragments and PBMCs. Many kinds of antigens and pathogens were used for the analysis.”

Comment 4: Formatting needs to be corrected in page 6, line 253-254

Response: We appreciate the reviewer’s comment. We corrected the format.

Reviewer 2 Report

This manuscript was titled as “Humanized Mice as an Effective Evaluation System for Peptide Vaccines and Immune Checkpoint Inhibitors”, and written about “Difficulties in the Development of Peptide Vaccines” in part 2, “Immune Checkpoint Inhibitors and Reagents for Side-Effect Regulation” in part 3, and “Humanized Mouse Models for the Evaluation of the Human Immune Environment” in part 4. Although this review was written well, there are some concerns as below.

Comments

The main intendment of this article might be “Humanized Mice Model as an Effective Evaluation System to predict the efficacy and adverse events of vaccination”, so the authors should make some figures which clearly present the mice system described in this manuscript, how to and for what immune system is constructed, of what immunity can be examined, and useful for what diseases or selection of treatment.

Cellular immune responses might be more important for cancer therapy, and the manuscript was written about cancer therapy as much as infection. However, there is little description on the cellular immune system of humanized mice. The authors should added the descriptions about humanized mice models on cellular immune system.

Page 2 Line 69: they described that “immunogenic peptide with more than 8 to 10 residues which can be further presented by the patient MHC (class I for cytotoxic T-cell activation and class II for antibody production)”. However, it is generally known “long peptides” are needed for class II induction.

Page 7 Line 277: they described that “Although HLA class I transgenic mice evoke partially improved human humoral immune responses”. However, HLA class I transgenic mice like HLA-A24 or A02 transgenic mice can induce cellular immune responses as presented by ELISPOT assay or cytotoxic assay.

Author Response

Response to the comments of Reviewer #2

Comment 1: The main intendment of this article might be “Humanized Mice Model as an Effective Evaluation System to predict the efficacy and adverse events of vaccination”, so the authors should make some figures which clearly present the mice system described in this manuscript, how to and for what immune system is constructed, of what immunity can be examined, and useful for what diseases or selection of treatment.

Response: We appreciate the reviewer’s comment. According to the suggestion, we added a Figure which explain the humanized mouse model in page 7 after line 247 with figure legend “Figure 1. Three strategies for the reconstitution of human immunity in the immunodeficient mouse. The transplanted tissues are, HSC, Lymphoid tissues or the fragments and PBMCs. Many kinds of antigens and pathogens were used for the analysis.”

Comment 2: Cellular immune responses might be more important for cancer therapy, and the manuscript was written about cancer therapy as much as infection. However, there is little description on the cellular immune system of humanized mice. The authors should added the descriptions about humanized mice models on cellular immune system.

Response: We appreciate the reviewer’s comment. As the reviewer pointed out, cellular immune system is more important for cancer therapy and the aim of humanized mouse system tended to evaluate the cellular immunity. Based on the situation, there are good reviews previously published. However, the importance of humoral immunity has not been shed light on. Therefore, we would like to emphasize the humoral immune system which may have some important role in the cancer immunity. However, we accept the advice to mention about the cellular immune system of humanized mouse and added the description asAmong them, HLA class I transgenic mice evoked antigen-specific cytotoxic T cell response against HSV virion protein peptide [130] or WT1 peptide [131]. The success of the reconstitution of human cellular immune response is followed by an adoptive transfer therapy model using humanized mouse system [132]. Consequently, the established patient-derived xenograft (PDX) system which transplants patient’s cancer tissues (minimal standard was reported by Meehan et al. [133]) combined with patient’s T cells is widely accepted. The detail was intensively reviewed by other researchers [134-136]. On the other hand, the response of HLA class II transgenic mice did not completely mimic the human humoral immunity [137-139].” in page 9 line 293-301 and after the description of already mentioned anti-cancer effect, added the description in more detail as “The mouse system had an advantage that the restriction of HLA type could be avoided by using PBMC, which contain the same patients’ T cells and antigen presenting cells. However, they did not detect anti-cancer antibody production in this study.” in line 306 to 309. We also explained as “Moreover, PBMC engrafts in the mice and can reconstitute human cellular immunity. However,” in page 10 line 351-352.

Comment 3: Page 2 Line 69: they described that “immunogenic peptide with more than 8 to 10 residues which can be further presented by the patient MHC (class I for cytotoxic T-cell activation and class II for antibody production)”. However, it is generally known “long peptides” are needed for class II induction.

Response: We apologize that the description was not adequate for the explanation. We meant that the peptide was longer than 8 or 10 amino acids, which meant that the peptide is much more longer than 8 amino acids. We changed the description as “more than 8 up to 30” in page 2 line 71.

Comment 4: Page 7 Line 277: they described that “Although HLA class I transgenic mice evoke partially improved human humoral immune responses”. However, HLA class I transgenic mice like HLA-A24 or A02 transgenic mice can induce cellular immune responses as presented by ELISPOT assay or cytotoxic assay.

Response: We appreciate the reviewer’s comment. The comment is related to the comment 2. As reviewer commented, these mice were reported to induce cellular immune response. Therefore, we involved the discussion of this issue to the description of cellular immunity of humanized mouse we mentioned in the response to Comment 2. (in page 9 line 293-301)

Round 2

Reviewer 2 Report

 I consider that the manuscript is revised properly in response to the reviewers' comments.